# IgG Antibody Responses and Immune Persistence of Two Doses of BBIBP-CorV Vaccine or CoronaVac Vaccine in People Living with HIV (PLWH) in Shenzhen, China

**DOI:** 10.3390/vaccines10060880

**Published:** 2022-05-31

**Authors:** Guang Zeng, Liumei Xu, Shuidong Feng, Jie Tang, Xiaohui Wang, Guilian Li, Yongxia Gan, Chenli Zheng, Jin Zhao, Zhengrong Yang

**Affiliations:** 1School of Public Health, University of South China, Hengyang 421001, China; 20202014210997@stu.usc.edu.cn (G.Z.); 1998000341@usc.edu.cn (S.F.); 20192014210823@stu.usc.edu.cn (J.T.); 2Department of HIV/STDS Prevention and Control, Shenzhen Center for Disease Prevention and Control, Shenzhen 518055, China; wangxh@szcdc.net (X.W.); gracili8761@gmail.com (G.L.); ganhaha@szcdc.net (Y.G.); chenli9700@hotmail.com (C.Z.); zhaoj@szcdc.net (J.Z.); 3The Third People’s Hospital of Shenzhen, Shenzhen 518020, China; he2020@szsy.sustech.edu.cn

**Keywords:** PLWH, BBIBP-CorV vaccine, CoronaVac vaccine, S-RBD-IgG, CD4^+^T cell, viral load, immune persistence

## Abstract

The purpose of this study was to preliminarily evaluate the immunogenicity and immune persistence of inactivated SARS-CoV-2 vaccines in PLWH in the real world. We collected blood samples from 132 PLWH aged 18–59 years who were vaccinated with two doses of BBIBP-CorV vaccine (Sinopharm) or CoronaVac vaccine (SinoVac) at 28 ± 7 days and 180 ± 20 days the after second dose, to detect the level of Spike receptor binding domain-protein specific IgG (S-RBD-IgG) by using chemiluminescence. We found that the BBIBP-CorV vaccine or the CoronaVac vaccine induced lower S-RBD-IgG antibody seropositivity rates and levels in PLWH than in healthy controls (HCs). The BBIBP-CorV vaccine or the CoronaVac vaccine induced lower humoral immune responses in PLWH, having lower CD4^+^T cell counts (<350 cells/μL) compared to PLWH, and having higher CD4^+^T cell counts (≥350 cells/μL) after a second dose of vaccination. The BBIBP-CorV vaccine or the CoronaVac vaccine induced lower S-RBD-IgG antibody levels in PLWH, having CD4^+^T cell counts ≥350 cells/μL compared to HCs. No negative effects were observed in terms of the CD4^+^T cell counts and HIV RNA viral load (VL) of PLWH after vaccination. Ninety-nine PLWH and eighty-three HCs completed a second blood collection for testing; we found a statistically significant decrease in the humoral immune response both in PLWH and HCs from 28 days to 180 days after a second dose of BBIBP-CorV vaccine or CoronaVac vaccine. The S-RBD-IgG antibody induced by the BBIBP-CorV vaccine or the CoronaVac vaccine declined faster in the PLWH population than in the healthy population, and two doses of the BBIBP-CorV vaccine or the CoronaVac vaccine may not be enough to provide PLWH with persistent immunity against SARS-CoV-2. It is necessary for PLWH to be prioritized for a third dose over the healthy population, but the immunogenicity of the third dose of the homologous or heterologous vaccine requires further study.

## 1. Introduction

Individual vaccination with SARS-CoV-2 vaccines approved by the World Health Organization for emergency use can effectively reduce the prevalence of severe cases and deaths. Widespread vaccination is essential to alleviate the global pandemic of Coronavirus Disease 2019 (COVID-19). In China, people are vaccinated with two inactivated SARS-CoV-2 vaccines widely (BBIBP-CorV vaccine and CoronaVac vaccine), and clinical trials have shown satisfactory immunogenicity in the general population [1,2]. A previous study showed that even for people with compromised immune systems, including PLWH, the overall benefits of vaccination with SARS-CoV-2 vaccines outweigh the potential risks [3]. The Centers for Disease Control and Prevention of America recommended that PLWH should be fully vaccinated with SARS-CoV-2 vaccines regardless of CD4^+^T cell counts or HIV RNA viral loads (VLs) [4]. Based on the theoretical low risk and potential high benefit of inactivated SARS-CoV-2 vaccines, on 3 April 2021, the AIDS and Hepatitis C Professional Group recommended that PLWH with well-suppressed VLs should be vaccinated with inactivated SARS-CoV-2 vaccines as soon as possible [5]. However, so far, studies evaluating the immunogenicity of inactivated SARS-CoV-2 vaccines in PLWH are extremely limited, and studies on immune persistence are also lacking.

In this study, peripheral venous blood was collected at 28 days and 180 days after PLWH and HCs were vaccinated with two doses of BBIBP-CorV vaccine or CoronaVac vaccine, for the detection of S-RBD-IgG antibody seropositivity rates and antibody levels. We compared the immunogenicity and immune persistence of two-dose inactivated SARS-CoV-2 vaccines in PLWH and HCs in the real world and explored the effect of CD4^+^T cell counts before the first dose of vaccine on the humoral immune response. In addition, we explored whether vaccination with inactivated SARS-CoV-2 vaccines affects CD4^+^T cell counts and plasma VLs in PLWH.

## 2. Materials and Methods

### 2.1. Object Recruitment

The Third People’s Hospital of Shenzhen is a designated hospital for AIDS treatment in Shenzhen. PLWH who receive routine follow-up come to the hospital every 3 months to have their CD4^+^T cell counts and VLs measured and receive a 3-month course of antiretroviral therapy (ART) drugs. PLWH who were on routine follow-up and who had completed two doses of BBIBP-CorV vaccine or CoronaVac vaccine were recruited by convenience sampling. Inclusion criteria: (1) age 18–59 years old; (2) no history of SARS-CoV-2 infection nor history of close or indirect contact with people diagnosed with COVID-19; and (3) cooperation with questionnaire survey, blood collection, and follow-up. Healthy controls (HCs) were then recruited through internet advertisement. This study complies with the requirements of the Declaration of Helsinki and was approved by the Ethics Committee of Shenzhen Center for Disease Control and Prevention (No. QS2021070043, 10 August 2021). All the participants signed a written informed consent. All the collected information was anonymized and kept confidential.

### 2.2. Questionnaire Survey

With the assistance of the investigators, PLWH completed the questionnaire, which consisted of three items: (1) demographic data; (2) HIV characteristics; and (3) the manufacturer of the inactivated SARS-CoV-2 vaccines and the date of each dose of vaccination. Demographic data included sex and age; HIV characteristics data included the mode of HIV transmission and duration of ART treatment before the first dose of vaccine. CD4^+^T cell counts and VLs were measured before the first dose and after the second dose of inactivated SARS-CoV-2 vaccines.

### 2.3. Blood Collection and Testing

Blood was collected longitudinally from PLWH and HC after informed consent was obtained. A total of 5 mL of peripheral venous blood was collected at 28 days and 180 days after the second dose of BBIBP-CorV vaccine or CoronaVac vaccine. We used magnetic particle chemiluminescence kits (Shengxiang Biotechnology, Changsha, China) to detect S-RBD-IgG in blood samples. The S-RBD-IgG antibody in the serum sample and the components in the reagent formed a complex of alkaline phosphatase-labeled antibody, S-RBD-IgG antibody, recombinant antigen, and magnetic powder. After the addition of the substrate, the alkaline phosphatase in the complex catalyzed the fluorescence of the substrate. The relative luminescence unit (RLU) of the complex to the substrate can indirectly reflect the level of S-RBD-IgG antibody. The detection result was expressed as the ratio of the sample luminescence value (S) to the cutoff (CO) (S/CO value), and the S/CO value was used as an indirect indicator of antibody level; with an S/CO value ≥ 1.0 being defined as S-RBD-IgG seropositivity.

### 2.4. Statistical Analysis

The data were organized using Microsoft Excel 2010, and statistical analysis was performed with IBM SPSS Statistics 25.0 (IBM Corp, Armonk, NY, USA) and GraphPad Prism 8.4.2 (GraphPad Software, La Jolla, CA, USA). Baseline characteristics were described by the using mean and standard deviation (SD) or median (interquartile ranges, IQR). Enumeration data were analyzed by constituent ratios or rates with 95% confidence intervals (95%CI). Antibody level S/CO values were described as the median (IQR). The Student’s t-test or nonparametric test was used to compare the difference in the mean or median between the two groups, and the chi-square test was used to compare the difference in the constituent ratio or rate between the two groups. For two-sided tests, a *p*-value of 0.05 or lower was considered statistically significant.

## 3. Results

### 3.1. Demographics, HIV Characteristics and Types of Vaccinations of PLWH

A total of 132 PLWH who had been vaccinated with two doses of inactivated SARS-CoV-2 vaccines were recruited, of whom 65 were vaccinated with BBIBP-CorV vaccine and 67 were vaccinated with CoronaVac vaccine. A total of 132 valid questionnaires were recovered. The majority of the ePLWH were male, with 119 patients (90.2%); the average age of PLWH was (34.05 ± 8.37) years old, the median age was 32 years old (IQR, 28–39), and the 30–39 age group accounted for the greatest proportion (46.2%). Most PLWH (61.4%) had an education level of high school and below; both the proportion of unmarried (72.7%) and homosexual sexual transmission (67.4%) PLWH were over two-thirds. Before the first dose of vaccine, the vast majority of the PLWH (87.9%) had received stable ART for more than 6 months, and 90.9% had undetectable plasma VLs (<50 copies/mL). Nearly half of the PLWH (49.2%) had CD4^+^T cell counts above 500 cells/µL. Six PLWH had not started ART before their first dose of vaccine, and their plasma VLs and CD4^+^T cell counts were unknown; two of them received the BBIBP-CorV vaccine and four of them received the CoronaVac vaccine. The median duration of stable ART in the 126 PLWH was 3.64 years (IQR, 1.33–5.76), and the median CD4^+^T cell count was 505 cells/µL (IQR, 385–658) before the first dose of vaccine. Detailed results are presented in Table 1.

### 3.2. The S-RBD-IgG Antibody Data of First Blood Collection at 28 ± 7 Days after PLWH and HC Received Two Doses of BBIBP-CorV Vaccine or CoronaVac Vaccine

In order to study the humoral immune response of PLWH vaccinated with two doses of the BBIBP-CorV vaccine or CoronaVac vaccine, we recruited 65 HC and 67 HC vaccinated with two doses of the BBIBP-CorV vaccine or CoronaVac vaccine at a ratio of 1:1, respectively; two HCs vaccinated with the CoronaVac vaccine withdrew from blood collection halfway. There were no statistically significant differences between the PLWH and HCs vaccinated with the BBIBP-CorV vaccine or the CoronaVac vaccine in terms of age, gender, the interval between the two doses, and the time of blood collection after the second dose of vaccine (all *p* > 0.05, Table 2).

A total of 28 days after the second dose of the BBIBP-CorV vaccine, the S-RBD-IgG antibody seropositivity rate of PLWH was 81.5% (53/65), while that of the HCs was 93.8% (61/65), the difference was statistically significant (*p* = 0.033). The median S/CO values of S-RBD-IgG antibody in PLWH and the HCs were 2.5 (IQR, 1.2–7.4) and 11.4 (IQR, 6.6–15.6), respectively, and the difference was statistically significant (*p* < 0.001). In the PLWH group, the numbers of participants with S/CO values <1, 1–9, 10–19, and ≥20 were 12 (18.5%), 42 (64.6%), 7 (10.8%), and 4 (6.2%), respectively; the numbers of participants with S/CO values <1, 1–9, 10–19, and ≥20 were 4 (6.2%), 21 (32.3%), 33 (50.8%), and 7 (10.8%) in the HC group, respectively; a statistical difference was observed in the distribution of antibody levels between the PLWH group and HC group (*p* < 0.001).

A total of 28 days after the second dose of the CoronaVac vaccine, the S-RBD-IgG antibody seropositivity rates were 86.6% (58/67) and 96.9% (63/65) in the PLWH group and the HC group, respectively; the difference was statistically significant (*p* = 0.031). The median S/CO values of S-RBD-IgG antibody in the PLWH and HC groups were 7.0 (IQR, 2.8–13.3) and 12.7 (IQR, 8.6–16.2), respectively, and the difference was statistically significant (*p* < 0.001). In the PLWH group, the numbers of participants with S/CO values <1, 1–9, 10–19, and ≥20 were 9 (13.4%), 33 (49.3%), 17 (24.4%), and 8 (11.9%), respectively; the numbers of participants with S/CO values <1, 1–9, 10–19, and ≥20 were 2 (3.1%), 18 (27.7%), 36 (55.4%), and 9 (13.8%) in the HC group, respectively; a statistical difference was observed in the distribution of antibody levels between the PLWH group and HC group (*p* = 0.001) (Table 3).

### 3.3. Influence of CD4^+^T Cell Counts before First Dose of Vaccine on S-RBD-IgG Antibody Responses at 28 ± 7 Days after PLWH Vaccination with Two Doses of BBIBP-CorV Vaccine or CoronaVac Vaccine

We further evaluated the humoral immune responses of two doses of BBIBP-CorV vaccine or CoronaVac vaccine on PLWH with different immune states. According to the CD4^+^T cell counts before the first dose of vaccine, PLWH were divided into two groups: CD4^+^T cell counts <350 cells/µL and CD4^+^T cell counts ≥350 cells/µL. In the PLWH vaccinated with the BBIBP-CorV vaccine, the S-RBD-IgG antibody seropositivity rates of the two groups were 60.0% (9/15) and 87.5% (42/48), respectively; the difference was statistically significant (*p* = 0.046). The median S/CO values of S-RBD-IgG antibody in the two groups were 1.2 (IQR, 0.1–2.3) and 3.2 (IQR, 1.4–8.0), respectively; a statistical difference was also observed (*p* = 0.004). Therefore, the BBIBP-CorV vaccine was likely to induce lower humoral immune responses in PLWH, with lower CD4^+^T cell counts (<350 cells/µL).

In the PLWH vaccinated with the CoronaVac vaccine, the S-RBD-IgG antibody seropositive rates of the two groups were 63.6% (7/11) and 92.3% (48/52), respectively; the difference was statistically significant (*p* = 0.036). The median S/CO values of S-RBD-IgG antibody were 4.2 (IQR, 0.7–9.1) and 7.9 (IQR, 4.1–13.4), respectively; and the difference was statistically significant (*p* = 0.020). Likewise, the CoronaVac vaccine was likely to induce lower humoral immune responses in PLWH, having lower CD4^+^T cell counts (<350 cells/µL) (Figure 1A,B).

### 3.4. The S-RBD-IgG Antibody Data Induced by Two Doses of BBIBP-CorV Vaccine or CoronaVac Vaccine in PLWH Having CD4^+^T Cell Counts ≥350 Cells/µL before First Dose of Vaccine and Healthy Controls

We further evaluated the humoral immune responses to two doses of the BBIBP-CorV vaccine or CoronaVac vaccine in PLWH with CD4+T cell counts ≥350 cells/µL and HC. There were no significant differences in the S-RBD-IgG antibody seropositivity rates induced by the BBIBP-CorV vaccine (87.5% vs. 93.8%) or the CoronaVac vaccine (92.3% vs. 96.9%) between PLWH having CD4+T cell counts ≥350 cells/µL and the HCs (Figure 1C). In the participants vaccinated with the BBIBP-CorV vaccine, the median S/CO values of S-RBD-IgG antibody in the two groups were 3.2 (IQR, 1.4–8.0) and 11.4 (IQR, 6.6–15.6), respectively; a statistical difference was observed (*p* < 0.001). In the participants vaccinated with the CoronaVac vaccine, the median S/CO values of S-RBD-IgG antibody in the two groups were 7.9 (IQR, 4.1–13.4) and 12.7 (IQR, 8.6–16.2), respectively; a statistical difference was also observed (*p* = 0.012) (Figure 1D).

### 3.5. Effects of Vaccination with BBIBP-CorV Vaccine or CoronaVac Vaccine on VLs and CD4^+^T Cell Counts of PLWH

To determine the effect of PLWH vaccination with the BBIBP-CorV vaccine or the CoronaVac vaccine on their VLs and CD4^+^T cell counts, we collected follow-up data on VLs and CD4^+^T cell counts within 4 weeks before the first dose of vaccine and 3–5 weeks after the second dose of vaccine. The CD4^+^T cell counts of PLWH vaccinated with the BBIBP-CorV vaccine was 499 (IQR, 357–688) cells/µL before the first dose and increased to 514 (IQR, 381–683) cells/µL after the second dose, but the difference was not statistically significant (*p* = 0.523). The CD4^+^T cell counts of PLWH vaccinated with the CoronaVac vaccine increased from 511 (IQR, 416–643) cells/μL before the first dose to 529 (IQR, 398–609) cells/µL after the second dose; no statistical difference was observed (Figure 1E). No VLs rebound was observed in either group and the rates of undetectable VLs (<50 copies/mL) increased (all 95.2% vs. 100%), although the difference was not statistically significant (Figure 1F). It was suggested that PLWH had no adverse effect on VLs and CD4^+^T cell counts after vaccination with two doses of the BBIBP-CorV vaccine or the CoronaVac vaccine.

### 3.6. S-RBD-IgG Antibody Data of Second Blood Collection at 180 ± 20 Days after PLWH and HCs Received Two Doses of BBIBP-CorV Vaccine or CoronaVac Vaccine

In order to study the immune persistence of two doses of inactivated SARS-CoV-2 vaccines in PLWH and HCs, we collected blood from PLWH and HCs at 180 ± 20 days after the second dose of the BBIBP-CorV vaccine or the CoronaVac vaccine for S-RBD-IgG antibody detection. Among the PLWH vaccinated with the BBIBP-CorV vaccine, forty-nine PLWH and 47 HCs completed the second blood collection, respectively; among the PLWH vaccinated with the CoronaVac vaccine, 50 PLWH and 36 HCs completed the second blood collection, respectively. Their characteristics are as follows (Table 4). All the participants had never been infected with SARS-CoV-2. The S-RBD-IgG antibody seropositivity rate of PLWH vaccinated with the BBIBP-CorV vaccine decreased from 79.6% (39/49) at 28 days after the second dose to 16.3% (8/49) at 180 days after the second dose (the decrease in the seropositivity rate was statistically significant (*p* < 0.001)), while that of the HCs decreased from 93.6% (44/47) to 38.3% (18/47) (the difference was also statistically significant (*p* < 0.001)). Meanwhile, there was a statistically significant difference in the S-RBD-IgG antibody seropositivity rates between the PLWH and the HCs at 180 days (16.3% vs. 38.3%, *p* = 0.015) (Figure 1G). The median S/CO value of S-RBD-IgG antibody decreased from 2.3 (IQR, 1.1–6.8) to 0.2 (IQR, 0.1–0.5) in PLWH while that of the HCs decreased from 11.4 (IQR, 7.3–16.7) to 0.8 (IQR, 0.4–1.6). The decrease in antibody levels was statistically significant in the PLWH and HCs, respectively (all *p* < 0.001). There was a statistically significant difference in the S-RBD-IgG antibody median S/CO values between the PLWH and the HCs at 180 days (0.2 (IQR, 0.1–0.5) vs. 0.8 (IQR, 0.4–1.6), *p* < 0.001) (Figure 1H).

The S-RBD-IgG antibody seropositivity rate of PLWH vaccinated with the CoronaVac vaccine decreased from 84.0% (42/50) at 28 days after the second dose to 26.0% (13/50) at 180 days after the second dose (the decrease of the seropositive rate was statistically significant (*p* < 0.001)), while that of HCs decreased from 97.2% (35/36) to 52.8% (19/36) (the difference was also statistically significant (*p* < 0.001)). Meanwhile, there was a statistically significant difference in the S-RBD-IgG antibody seropositivity rates between the PLWH and the HCs at 180 days (26.0% vs. 52.8%, *p* = 0.011) (Figure 1I). The median S/CO value of S-RBD-IgG antibody decreased from 7.0 (IQR, 2.7–12.7) to 0.6 (IQR, 0.4–1.0), while that of the HCs decreased from 12.4 (IQR, 9.0–16.9) to 1.3 (IQR, 0.8–2.3). The decrease in antibody levels was statistically significant in PLWH and HCs, respectively (all *p* < 0.001). There was a statistically significant difference in the S-RBD-IgG antibody median S/CO values between PLWH and HCs at 180 days (0.6 (IQR, 0.4–1.0) vs. 1.3 (IQR, 0.8–2.3), *p* < 0.001) (Figure 1J). Therefore, it is suggested that the S-RBD-IgG antibody induced by the BBIBP-CorV vaccine or the CoronaVac vaccine declined faster in the PLWH population than in the healthy population.

## 4. Discussions

The Centers for Disease Control and Prevention of America pointed out that immunocompromised populations such as PLWH were not only at an increased risk of severe COVID-19 disease and death, but that their immune response to SARS-CoV-2 vaccines may not be as strong as that of healthy individuals [6]. Our study found that the S-RBD-IgG antibody seropositivity rates and antibody levels were lower in PLWH at 28 ± 7 days after two doses of the BBIBP-CorV vaccine or the CoronaVac vaccine than in the HCs, which was consistent with the findings of previous studies [7,8,9]. The structural alterations in the germinal center of PLWH may affect T follicular helper cell and germinal center B cell homeostasis, thereby preventing the elevation of effective humoral immunity [10]. Therefore, humoral immune responses between PLWH and healthy individuals are generally comparable. However, previous studies have shown that after two doses of BBIBP-CorV vaccine or CoronaVac vaccine, PLWH and HCs could produce similar S-RBD-IgG titers [11,12]. This supports the conclusion that PLWH can be vaccinated with inactivated SARS-CoV-2 vaccines.

The CD4^+^T cell count was considered to reflect the immune status of PLWH, and an effective response to a vaccine requires appropriate CD4^+^T cell function to orchestrate the immune response [13]. We found that the BBIBP-CorV vaccine or the CoronaVac vaccine induced a significantly higher S-RBD-IgG antibody seropositivity rate and antibody level in PLWH with CD4^+^T cell counts ≥350 cells/µL than those of PLWH with CD4^+^T cell counts <350 cells/µL before the first dose of the vaccine. It was suggested that the humoral immune response of inactivated SARS-CoV-2 vaccines in PLWH may be related to CD4^+^T cell counts, which was consistent with the finding of Liu [12]. We further found that although there was no statistical difference in the S-RBD-IgG antibody seropositivity rate between PLWH with CD4^+^T cell counts ≥350 cells/µL and HCs, the S-RBD-IgG antibody level of PLWH with CD4^+^T cell counts ≥350 cells/µL was significantly lower than that of the HCs. In previous studies, PLWH with CD4^+^T cell counts <500 cells/µL had a lower immunogenicity than healthy controls [8]. Strategies should be developed to improve the immunogenicity induced by vaccines for PLWH, especially for PLWH having lower CD4^+^T cell counts; vaccines with higher antigen titers or booster doses for PLWH may be needed. Frater found that the ChAdOx1nCoV-19 vaccine induced no attenuation of humoral immune responses in PLWH with a CD4^+^T cell count >350 cells/μL [14]. Therefore, determining the type of immunization for SARS-CoV-2 vaccines based on CD4^+^T cell counts may be important to improve humoral immune responses in PLWH.

CD4^+^T cell counts and VLs are specific indicators to assess the efficacy of ART, and the effect of vaccinating PLWH with inactivated SARS-CoV-2 vaccines is unclear. Our results show that PLWH vaccinated with inactivated SARS-CoV-2 vaccines exhibited no negative effects on their CD4^+^T cell counts and VLs, which was consistent with the findings of a previous study [12]. We found that all PLWH had undetectable VLs at 28 days after the second dose of the BBIBP-CorV vaccine or the CoronaVac vaccine, but the effect of stable ART could not be excluded. A previous study pointed out that the VLs of PLWH was significantly decreased after vaccination with two doses of the BBIBP-CorV vaccine [11], and another study reported that the VLs of PLWH increased after vaccination with the mRNA-1273 vaccine [15]. The mechanism of VL fluctuation is not clear, and whether the type of vaccine affects the VLs of PLWH warrants further study.

Previous studies found that the S-RBD-IgG antibody level of recipients vaccinated with two doses of the BBIBP-CorV vaccine continued to decline over time. The antibody level began to decline three months after vaccination [16,17], the average decline level remained at 1.71 AU/mL or 1.66 BAU per day [18] and disappeared five months after vaccination [19]. The decrease in antibody concentration has raised concerns about the immune persistence induced by vaccination. Our study is the first to report the immune persistence of PLWH after two doses of the BBIBP-CorV vaccine or the CoronaVac vaccine. We found a statistically significant decrease in the seropositivity rates and levels of the S-RBD-IgG antibody both in PLWH and HCs from 28 days to 180 days after the second dose of the vaccine (all *p* < 0.001). In our study, the S-RBD-IgG antibody seropositivity rate at 180 days in PLWH was not only significantly lower than that of the HCs, but also lower than that of healthy adults in a previous study (26.0% vs. 35.2%) [20]. This may be related to the faster decline in vaccine-induced antibodies in PLWH than in healthy people [21]. Likewise, the S-RBD-IgG antibody level at 180 days in PLWH was significantly lower than that of the HCs, which suggests that two doses of the BBIBP-CorV vaccine or the CoronaVac vaccine may not be enough to provide PLWH with long-term immunity against SARS-CoV-2. A previous study pointed out that the S-RBD-IgG antibody level of the immunocompromised population vaccinated with the BNT162b2 vaccine decreased more significantly over time [22]. Therefore, it is necessary for PLWH to be prioritized for a third dose over the healthy population. Three doses of the BBIBP-CorV vaccine [23,24] or the CoronaVac vaccine [20,25] can significantly increase the antibody level and duration of protection in healthy adults. The immunogenicity of a third dose of homologous or heterologous vaccine after receiving two doses of inactivated SARS-CoV-2 vaccines in PLWH requires further study.

Our study has several limitations. First, at the time of this study, Chinese policy mandated that adults aged 18–59 were vaccinated, so the PLWH in this study were aged between 18 and 59; therefore, the immunogenicity and immune persistence of inactivated SARS-CoV-2 vaccines in PLWH under 18 years and over 60 years requires further study. Second, the proportion of male PLWH was very high. A previous study found that the BBIBP-CorV vaccine induced the highest S-RBD-IgG antibody concentrations in healthy young individuals and women [16]. Whether the intensity of the humoral immune response induced by two doses of inactivated SARS-CoV-2 vaccines in PLWH is affected by age and gender requires further study. Third, the sample size of this study was relatively small, and the humoral immune responses induced by two doses of inactivated SARS-CoV-2 vaccines in PLWH need to be validated in a cohort study with a larger sample.

## 5. Conclusions

In summary, the S-RBD-IgG antibody seropositivity rate and antibody level were lower in PLWH at 28 ± 7 days after two doses of the BBIBP-CorV vaccine or the CoronaVac vaccine than in HCs. The BBIBP-CorV vaccine or the CoronaVac vaccine induced lower humoral immune responses in PLWH having lower CD4^+^T cell counts (<350 cells/uL) compared with PLWH having higher CD4^+^T cell counts (≥350 cells/uL) after the second dose of vaccine. The BBIBP-CorV vaccine or the CoronaVac vaccine induced lower S-RBD-IgG antibody levels in PLWH with CD4^+^T cell counts ≥350 cells/uL compared to the HCs. PLWH vaccinated with inactivated SARS-CoV-2 vaccines exhibited no negative effect on their CD4^+^T cell counts and VLs. The S-RBD-IgG antibody response induced by the BBIBP-CorV vaccine or the CoronaVac vaccine declined faster in the PLWH population than the healthy population; two doses of the BBIBP-CorV vaccine or the CoronaVac vaccine may not be enough to provide PLWH with persistent immunity against SARS-CoV-2. Therefore, PLWH should be prioritized for a third dose, in order to promote the well-being of this vulnerable population, but the immunogenicity of a third dose of homologous or heterologous vaccine after receiving two doses of inactivated SARS-CoV-2 vaccines in PLWH requires further study.

## Figures and Tables

**Figure 1 vaccines-10-00880-f001:**
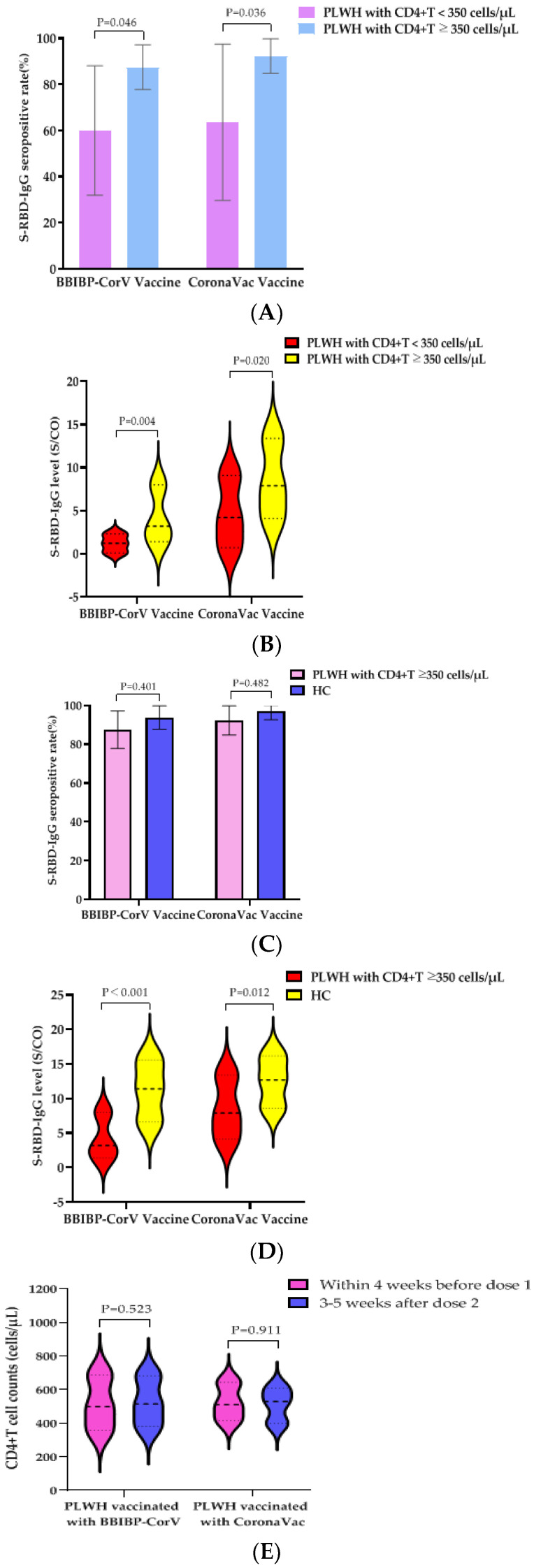
IgG antibody responses and immune persistence of two doses of BBIBP-CorV Vaccine or CoronaVac Vaccine in PLWH and HC. Twenty-six PLWH having CD4^+^T cell counts <350 cells/µL were vaccinated with BBIBP-CorV Vaccine (n = 15) or CoronaVac Vaccine (n = 11); one hundred PLWH having CD4^+^T cell counts ≥350 cells/µL were vaccinated with BBIBP-CorV Vaccine (n = 48) or CoronaVac Vaccine (n = 52); one hundred and thirty HCs were vaccinated with BBIBP-CorV Vaccine (n = 65) or CoronaVac Vaccine (n = 65). Ninety-nine PLWH were vaccinated with BBIBP-CorV Vaccine (n = 49) or CoronaVac Vaccine (n = 50) and completed second blood collection at 180 days after second dose; eighty-three HCs were vaccinated with BBIBP-CorV Vaccine (n = 47) or CoronaVac Vaccine (n = 36) and completed second blood collection at 180 days after second dose. (**A**) The S-RBD-IgG antibody seropositivity rates of PLWH with different CD4^+^T cell counts at 28 days after vaccination. (**B**) The S-RBD-IgG antibody levels of PLWH with different CD4^+^T cell counts at 28 days after vaccination. (**C**) The S-RBD-IgG antibody seropositivity rates of PLWH having CD4^+^T cell counts ≥350 cells/µL and HCs at 28 days after vaccination. (**D**) The S-RBD-IgG antibody levels of PLWH having CD4^+^T cell counts ≥350 cells/µL and HCs at 28 days after vaccination. (**E**) Effects of vaccination with BBIBP-CorV vaccine or CoronaVac vaccine on CD4^+^T cell counts of PLWH. (**F**) Effects of vaccination with BBIBP-CorV vaccine or CoronaVac vaccine on VL of PLWH. (**G**) The S-RBD-IgG antibody seropositivity rates of PLWH and HCs vaccination with BBIBP-CorV vaccine at 28 and 180 days after the second dose. (**H**) The S-RBD-IgG antibody levels of PLWH and HCs vaccination with BBIBP-CorV vaccine at 28 and 180 days after the second dose. (**I**) The S-RBD-IgG antibody seropositivity rates of PLWH and HCs vaccination with CoronaVac vaccine at 28 and 180 days after the second dose. (**J**) The S-RBD-IgG antibody levels of PLWH and HCs vaccination with CoronaVac vaccine at 28 and 180 days after the second dose.

**Table 1 vaccines-10-00880-t001:** Demographics, HIV characteristics, and types of vaccination of PLWH (N = 132).

Variables		Statistic Value
Sex (n, %)	Male	119 (90.2%)
	Female	13 (9.8%)
Age (years, n, %)	20–29	43 (32.3%)
	30–39	61 (46.2%)
	40–49	21 (15.9%)
	50–59	7 (5.3%)
Mode of HIV transmission (n, %)	Homosexual sexual transmission	89 (67.4%)
	Heterosexual sexual transmission	42 (31.8%)
	Others	1 (0.8%)
ART time before first dose of vaccine	Not started	6 (4.5%)
(months, n, %)	≤6	10 (7.6%)
	>6	116 (87.9%)
VLs before the first dose of vaccine	<50	120 (90.9%)
(copies/mL, n, %)	≥50	6 (4.5%)
	Unknown	6 (4.5%)
CD4^+^T cell counts before first dose of vaccine (cells/µL, n, %)	0–199 200–349	7 (5.3%) 19 (14.4%)
	350–499	35 (26.5%)
	≥500	65 (49.2%)
	Unknown	6 (4.5%)
Types of vaccination (n, %)	BBIBP-CorV vaccine	65 (49.2%)
	CoronaVac vaccine	67 (50.8%)

**Table 2 vaccines-10-00880-t002:** PLWH and HCs’ characteristics of first blood collection after two doses of BBIBP-CorV vaccine or CoronaVac vaccine (N = 262).

Characteristics	PLWH ^a^ (n = 65)	HCs ^a^ (n = 65)	*p*	PLWH ^b^ (n = 67)	HCs ^b^ (n = 65)	*p*
Age (years, mean ± SD)	34.0 ± 8.6	34.3 ± 9.4	0.845	34.1 ± 8.3	34.6 ± 8.5	0.726
Male (n, %)	55 (84.6%)	55 (84.6%)	1.000	64 (95.5%)	60 (92.3%)	0.683
Interval between two doses (days, mean ± SD)	27.3 ± 6.0	28.0 ± 5.2	0.463	25.1 ± 4.3	25.0 ± 4.1	0.853
Blood collection after dose 2 (days, mean ± SD)	29.7 ± 3.4	28.9 ± 3.7	0.221	28.2 ± 3.7	28.0 ± 3.9	0.784

^a^ Vaccinated with BBIBP-CorV vaccine; ^b^ Vaccinated with CoronaVac vaccine.

**Table 3 vaccines-10-00880-t003:** The S-RBD-IgG antibody data for the first blood collection after PLWH and HCs received two doses of BBIBP-CorV vaccine or CoronaVac vaccine (N = 262).

Groups	N	S-RBD-IgG Antibody Seropositivity		S-RBD-IgG Antibody Levels				
		Numbers	Rate (95%CI) (%)	Median S/CO Value (IQR)	<1	1–9	10–19	≥20
PLWH ^a^	65	53	81.5 (71.8, 91.2)	2.5 (1.2–7.4)	12 (18.5%)	42 (64.6%)	7 (10.8%)	4 (6.2%)
HCs ^a^	65	61	93.8 (87.8, 99.8)	11.4 (6.6–15.6)	4 (6.2%)	21 (32.3%)	33 (50.8%)	7 (10.8%)
*p*			0.033	< 0.001	<0.001			
PLWH ^b^	67	58	86.6 (78.2, 94.9)	7.0 (2.8–13.3)	9 (13.4%)	33 (49.3%)	17 (24.4%)	8 (11.9%)
HCs ^b^	65	63	96.9 (92.6, 100.0)	12.7 (8.6–16.2)	2 (3.1%)	18 (27.7%)	36 (55.4%)	9 (13.8%)
*p*			0.031	<0.001		0.001		

^a^ Vaccinated with BBIBP-CorV vaccine; ^b^ Vaccinated with CoronaVac vaccine.

**Table 4 vaccines-10-00880-t004:** Characteristics of PLWH and HCs who had completed second blood collection after two doses of BBIBP-CorV vaccine or CoronaVac vaccine (N = 182).

Characteristics	PLWH ^a^ (n = 49)	HCs ^a^ (n = 47)	*p*	PLWH ^b^ (n = 50)	HCs ^b^ (n = 36)	*p*
Age (years, mean ± SD)	35.0 ± 9.0	33.2 ± 9.2	0.474	34.2 ± 7.9	35.2 ± 9.2	0.658
Male (n, %)	43 (87.8%)	42 (89.4%)	0.805	50 (100%)	34 (94.4%)	0.336
Interval between two doses (days, mean ± SD)	27.3 ± 6.3	27.8 ± 4.7	0.213	25.2 ± 4.6	25.2 ± 4.0	0.708
First blood collection after dose 2 (days, mean ± SD)	29.6 ± 3.7	29.2 ± 3.7	0.471	28.1 ± 4.0	27.7 ± 3.7	0.417
Second blood collection after dose 2 (days, mean ± SD)	183.3 ± 10.0	181.3 ± 8.8	0.281	179.6 ± 10.1	179.8 ± 7.4	0.793

^a^ Vaccinated with BBIBP-CorV vaccine; ^b^ Vaccinated with CoronaVac vaccine.

## Data Availability

The data presented in this study are available on request from the corresponding author. The data are not publicly available according to the ethical committee decision on the conduct of this study.

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
