# Peer review of "IgG Antibody Responses and Immune Persistence of Two Doses of BBIBP-CorV Vaccine or CoronaVac Vaccine in People Living with HIV (PLWH) in Shenzhen, China"

_vaccines, 2022, doi:10.3390/vaccines10060880_

Round 1
Reviewer 1 Report
The authors evaluate the immunogenicity and immune persistence of two different inactivated SARS-CoV-2 (CoV2) vaccines in people living with HIV (PLWH). The vaccines evaluated are BBIBP-CorV (Sinopharm) and CoronaVac (SinoVac). They compare immunogenicity between the two vaccines in PLWH and also compare their response against healthy controls who received the same vaccine regimens. Immunogenicity is evaluated ~28 day after the initial prime-boost series and persistence is evaluated ~180 days after the initial prime-boost series. Principle readouts are percentage seropositive and antibody titers (S/Co). The researchers also evaluated the impact of initial CD4 T cell counts on response to vaccination and whether vaccination impacted HIV disease progression (CD4 counts and viral loads).
The work is clearly presented and reasonably well written despite some language issues. Overall, the conclusions are clear and well supported by the data. Some improvements could be made as follows.
Major:
- The demographics section includes parameters such as “education level” and “marital status” that are irrelevant to the study and only serve to further stigmatize PLWH and is neither ethical nor necessary to include in the manuscript.
- While direct experimental comparisons are made with healthy controls at the 28 day post immunization time point, none is made at the 180 day timepoint. This precluded interpretation of whether immunity waned faster in this cohort of PLWH than healthy controls and missed an opportunity to make a valuable contribution to the question of whether PLWH may require boosters at earlier intervals than healthy individuals.
- In L251 it is unclear if the healthy controls in the referenced study only had 85.7% seroconversion. This should be clarified. Also, the reference should be moved to after the percentages are given.
- L254-255 the comment is contradictory to what is stated in L252-253. Please clarify.
- L267 their findings were “significantly” higher, not “relatively” higher. Please change.
Minor:
- Numerous typos, grammar, wording/language issues throughout. For example:
- L32 , L39 and L43 – I think “administer/administration” is the word required, not “vaccine”
- Final sentence in introduction is excessively long and needs to be split up.
- “Gender” is used when “sex” is the required word.
- Sentences should not start with numerals, write the number out.
- In more than one table “value” is misspelled as “vaule”
- “twice blood collection” is not the correct terminology/English – use “second” or simply refer to the timepoint.
- “pervious” instead of “previous”
- Starting sentences with “A Study” the s should not be capitalized
- Commas where a period is required
- The specific CLIA kit used should be specified in the M&M
- Figure 1B is miniscule and needs to be enlarged. Figure panel headers belong in the top left corner of each panel, not underneath.
Author Response
Dear Reviewer We appreciate the time and effort that you dedicated to providing feedback on our manuscript.Special thanks to you for your good comments and suggestions. Please see the attachment. Best regards
Reviewer 2 Report
Zeng et al studied the IgG antibody responses and immune persistence in HIV positive people, after they received two doses of the live attenuated vaccines – BBIBP-CorV vaccine and Corona Vac vaccine. They found that both live attenuated vaccines induced lower seropositive rate of the IgG antibody against SARS-CoV-2 spike-RBD in people living with HIV (PLWH) than in healthy controls. They further showed that both live attenuated vaccines induced lower humoral immune responses in PLWH having lower CD4+ T cell counts (<350 cells/uL) compared to PLWH having higher CD4+ T cell counts (≥350 cells/uL) after second dose of vaccination. The CD4+ T cell counts and viral load were not affected by two doses of vaccination in PLWH. Finally, they also found that the S-RBD-IgG seropositive rates and antibody levels dramatically declined after 180 days of second dose of vaccination in PLWH.
I have several comments as listed below.
Major comments:
- In table 4, the authors compared S-RBD-IgG antibody levels and the seropositive rate in PLWH having CD4+ T cells counts less than 350 cells/uL or more than 350 cells/uL. Have authors tried to compare PLWH having CD4+ T cell counts more than 350 cells/uL vs healthy controls? Is there any significant difference between these two groups?
- Table 6, in the discussion session, the authors did mention the S-RBD-IgG antibody level decay in general populations received two doses of BBIBP-CorV vaccine after 3 months of vaccination. However, it would be better to include the healthy control data in their own study, which can be presented in table 6.
Minor comments:
- Lines 20-22: write this sentence in a clearer way, it’s quite confusing. The authors probably would like to say such as “both live attenuated vaccines induced lower humoral immune responses in PLWH having lower CD4+ T cell counts (<350 cells/uL) compared to PLWH having higher CD4+ T cell counts (≥350 cells/uL) after second dose of vaccination”.
- Line 20: “health” control (HC) should be corrected to “healthy” control (HC)
- Line 25: correct two dose to two doses
- Figure 1: please make both figure A and B similar size
- Material and methods: 3.3. Blood collection and testing, this methodology was not clear for me.
- I would recommend the authors request the native speakers to edit the manuscript.
Author Response

(The authors gave the same response as above.)

Reviewer 3 Report
Title
IgG Antibody Responses and Immune Persistence of Two Doses of BBIBP-CorV Vaccine or CoronaVac Vaccine in People Living With HIV (PLWH) in Shenzhen, China
Abstract
The purpose of this study is to evaluate the immunogenicity and immune persistence of inactivated SARS-CoV-2 vaccines in PLWH in the real world preliminarily. We collected blood samples from 132 PLWH aged 18–59 years who vaccinated with two doses of BBIBP-CorV vaccine (Sinopharm) or CoronaVac vaccine (SinoVac) at 28 ± 7 days and 180 ± 20 days after second dose, to detect the level of Spike receptor binding domain-protein specific IgG (S-RBD-IgG) by using chemiluminescence. We found that BBIBP-CorV vaccine or CoronaVac vaccine induced lower S-RBD-IgG antibody seropositive rates and levels in PLWH than in health control (HC). BBIBP-CorV vaccine or CoronaVac vaccine induced relatively higher S-RBD-IgG antibody seropositive rate and antibody level in PLWH with CD4+T cell counts ≥ 350 cells/µL before first dose of vaccine. No negative effects were observed on the CD4+T cell counts and HIV RNA viral load (VL) of PLWH after vaccination with BBIBP-CorV vaccine or CoronaVac vaccine. 99 PLWH have completed blood collection twice for test; the S-RBD-IgG antibody seropositive rate of PLWH vaccinated with two dose of BBIBP-CorV vaccine decreased from 79.6% at 28 days after second dose to 16.3% at 180 days after second dose and the median S/CO value of S-RBD-IgG antibody decreased from 2.3 (IQR, 1.1-6.8) to 0.2 (IQR, 0.1-0.5); The S-RBD-IgG antibody seropositive rate of PLWH vaccinated with two doses of CoronaVac vaccine decreased from 84.0% at 28 days after second dose to 26.0% at 180 days after second dose and the median S/CO value of S-RBD-IgG antibody decreased from 7.0 (IQR, 2.7-12.7) to 0.6 (IQR, 0.4-1.0). It is suggested that BBIBP-CorV vaccine or CoronaVac vaccine may not be enough to provide PLWH with persistent immunity against SARS-CoV-2, and to vaccine a third booster dose as soon as possible is necessary, but the immunogenicity of the third dose of the homologous or heterologous vaccine requires further study.
I am afraid the manuscript has some intrinsic weakness :
- Convenience sampling was used. It is a non-probability sampling method, it is the most applicable and widely used method in clinical research. In this method, the investigators enroll subjects according to their availability and accessibility.
- We authors did not include stat method to compare the both groups have the equal baseline. This is evident from Table 1. Demographics, HIV characteristics and types of vaccines of PLWH (N=132).
Minor comment, some obvious typos, as shown below
- Materials and Methods
2.1. Object Recruitment
Blood was collected longitudinally from PLWH and HC after informed consent
WE used magnetic particle chemilumines
Author Response

(The authors gave the same response as above.)

Round 2
Reviewer 3 Report
All comments are well addressed.